# Exploring the Potential Hormonal Effects of Tire Polymers (TPs) on Different Species Based on a Theoretical Computational Approach

**DOI:** 10.3390/polym15071719

**Published:** 2023-03-30

**Authors:** Yu Wang, Hao Yang, Wei He, Peixuan Sun, Wenjin Zhao, Miao Liu

**Affiliations:** 1College of New Energy and Environment, Jilin University, Changchun 130012, China; 2College of Applied Chemistry and Materials, Zhuhai College of Science and Technology, Zhuhai 519041, China; 3MOE Key Laboratory of Resources Environmental Systems Optimization, North China Electric Power University, Beijing 102206, China

**Keywords:** tire polymers, toxicity, freshwater environment, marine environment, soil environment, molecular-dynamics approach

## Abstract

Tire polymers (TPs) are the most prevalent type of microplastics and are of great concern due to their potential environmental risks. This study aims to determine the toxicity of TPs with the help of molecular-dynamics simulations of their interactions with receptors and to highlight the differences in the toxicity characteristics of TPs in different environmental media (marine environment, freshwater environment, soil environment). For this purpose, five TPs—natural rubber, styrene–butadiene rubber (SBR), butadiene rubber, nitrile–butadiene rubber, and isobutylene–isoprene rubber—were analyzed. Molecular-dynamics calculations were conducted on their binding energies to neurotoxic, developmental, and reproductive receptors of various organisms to characterize the toxic effects of the five TPs. The organisms included freshwater species (freshwater nematodes, snails, shrimp, and freshwater fish), marine species (marine nematodes, mussels, crab, and marine fish), and soil species (soil nematodes, springtails, earthworms, and spiders). A multilevel empowerment method was used to determine the bio-toxicity of the TPs in various environmental media. A coupled-normalization method–principal-component analysis–factor-analysis weighting method—was used to calculate the weights of the TP toxicity (first level) categories. The results revealed that the TPs were the most biologically neurotoxic to three environmental media (20.79% and 10.57% higher compared with developmental and reproductive toxicity, respectively). Regarding the effects of TPs on organisms in various environmental media (second level), using a subjective empowerment approach, a gradual increase in toxicity was observed with increasing trophic levels due to the enrichment of TPs and the feeding behavior of organisms. TPs had the greatest influence in the freshwater-environment organisms according to the subjective empowerment approach employed to weight the three environmental media (third level). Therefore, using the minimum-value method coupled with the feature-aggregation method, the interval-deflation method coupled with the entropy-weighting method, and the standard-deviation normalization method, the three toxicity characteristics of SBR in three environmental media and four organisms were determined. SBR was found to have the greatest impact on the overall toxicity of the freshwater environment (12.38% and 9.33% higher than the marine and soil environments, respectively). The greatest contribution to neurotoxicity (26.01% and 15.95% higher than developmental and reproductive toxicity, respectively) and the greatest impact on snails and shrimp among organisms in the freshwater environment were observed. The causes of the heterogeneity of SBR’s toxicity were elucidated using amino-acid-residue analysis. SBR primarily interacted with toxic receptors through van der Waals, hydrophobic, π-π, and π-sigma interactions, and the more stable the binding, the more toxic the effect. The toxicity characteristics of TMPs to various organisms in different environments identified in this paper provide a theoretical basis for subsequent studies on the prevention and control of TMPs in the environment.

## 1. Introduction

Microplastics (MPs) are emerging pollutants and are widely present in various environmental media. Being hazardous to the environment and posing a risk to human health [1], they have attracted widespread attention worldwide. The particle size of MPs is usually <5 mm [2], and tire-rubber particles, asphalt, plastic fibers, plastic debris, and paint particles are defined as MPs [3,4]. Among them, tire polymers (TPs) produced by tire-wear particles are one of the most important MPs that have an environmental impact [5,6], and the TPs released by tire wear can reach up to 6.1 million tons/year [7]. Tire rubbers have a complex composition, with rubber polymers as the mainstay, enhanced by adding preservatives, antioxidants, plasticizers, and various additives [8,9]. It was shown previously that additives and monomers can also leach out of the particles and therefore are available to organisms. There are various types of tire rubbers, among which the two most commonly used types of rubber are styrene–butadiene rubber (SBR) and natural rubber (NR) [6]. SBR is the most commonly used rubber in automotive tires, and more than 70% of the SBR produced worldwide is used in tire manufacturing [10,11]. NR is the most commonly used rubber polymer in truck tires, accounting for 75% of global NR utilization [12]. Additionally, common tire rubbers include butadiene rubber (BR), nitrile–butadiene rubber (NBR), and isobutylene–isoprene rubber (IIR) [13,14,15,16,17]. Tire-rubber particles are released and accumulate during vehicle driving and braking of vehicles; the particles then flow into water bodies after being washed by rain and enter various environmental media [18,19]. The disposal of tire waste is currently being studied and the recycling of waste tires is currently divided into two main categories: physical and chemical [20]. The main physical disposal method is to recycle used tires, which can then be shredded and used as road-construction materials [21,22]. However, only 10% of the total amount of used tires can be recycled at present. Chemical disposal of used tires, including their use as biofuel, pyrolysis, chemical production, etc., may generate new pollution [23]. Therefore, the potential environmental hazards caused by used tires still need to be looked at, and there is an urgent need to develop safe and environmentally friendly means of disposing of used tires.

The initial TP studies associated with TPs mainly focused on analyzing the traffic and road environments [24,25]. An increasing number of studies have found that TPs are present in freshwater, marine, and soil environmental media worldwide; however, there are few studies on the influence of the interaction of multiple environmental media. Leads et al. [26] tested MPs in the estuarine tributaries of the Port of Charleston and found that blue fibers and TPs were the two most abundant MPs, among which TPs accounted for 17.1% of total MP content. Sieber et al. [27] calculated the cumulative amount of rubber MPs in soil and surface water in Switzerland from 1988 to 2018 and found that the cumulative amount of TPs in roadside soil and other soil could reach 170,000 and 100,000 tons in 30 years, respectively. Furthermore, a TP regression analysis found that 74% of the TPs were deposited in the roadside soil, 22% in surface water, and the remaining 4% in the soil 5 m away from the road. Goßmann et al. [28] detected TPs in crude and fine sea-salt samples, with a concentration range of 1–1815 μg/kg. The concentration of TPs in the salt of the Mediterranean Sea was higher than that in the salt of the Atlantic Ocean. Lenaker et al. [29] examined MPs in freshwater sources from the Milwaukee Estuary to Lake Michigan and the results showed that there was a concentration gradient of MPs with depth; the concentration of MPs with low density decreased with water depth, and MPs in sediments were mainly SBR particles. Müller et al. [30] collected mixed soil samples near German expressways and determined TPs in the soil samples by detecting the thermal-decomposition products of SBR. The concentration of TPs in those soil samples was as high as 155–15,898 mg/kg and TPs were mainly deposited in the surface soil within 2 m from the ground.

TPs have also been detected in animals. Nelms et al. [31] detected TPs such as nitrile rubber, neoprene rubber, ethylene-propylene–diene rubber, and SBR in the feces of Atlantic mackerel (*Scomber scombrus*) and gray seal (*Halichoerus grypus*). In general, researchers have focused on studying the toxic effects of TPs on organisms. However, few studies have focused on the extent of the toxic effects of TPs on various trophic levels of the food chain. In 2017, Pochron et al. [32] found that the growth rate of earthworms in soil contaminated with TP decreased by 14% after 33 days. In 2018, they also found that the weight of the earthworms that live in TP-contaminated soil was 14% lower than that of earthworms that live in uncontaminated soil. Moreover, the survival time of earthworms living in TP-contaminated soil decreased by 16.2%. Gualtieri et al. [33] tested the toxicity of TP eluates at 50 and 100 g/L on African clawed-toad embryos and found mortality rates of 80.2% and 26.8% and teratogenicity rates of 38.3% and 70.6%, respectively. To study an effect assessment of TPs in a freshwater environment, Cunningham et al. [34] selected zebrafish and large fleas, typical model organisms, for their experiments and found that TPs had teratogenic and lethal effects on zebrafish. In addition, the mortality of zebrafish exposed to nanoscale TPs was higher than that of zebrafish exposed to micron-scale TPs. The fatality rate reached 45%. Turner et al. [35] used tire-wear-particle leachate to simulate the toxic effect of TP on *Ulva lactuca* in a marine environment. The experimental results showed that the photochemical energy-conversion efficiency of *U. lactuca* exposed to TP leachate decreased non-linearly. Selonen et al. [10] studied the effects of TPs mixed with SBR, IIR, and NR on the survival and reproduction of soil invertebrates and found that exposure to mixed TPs decreased the reproduction rate of *Enterobacter faecalis* by up to 20% and the survival and reproduction rate of Candida by up to 38%. TPs can be ingested by lower-level organisms. However, there are few reports on the hazards of TPs to organisms at different trophic levels in each environment and on the variability of the toxic effects of different types of TPs on organisms.

Capolupo et al. [36] reported that TP leachates could cause neurotoxicity, reproductive toxicity, and developmental toxicity in organisms. This study investigates the toxicity of TPs in freshwater, seawater, and soil environments and the differences in toxicity between species. Freshwater nematodes [37], snails [38], shrimp [39], and freshwater fish [40] were selected as target organisms for freshwater environments. Marine nematodes [41], mussels [42], crabs [43], and marine fish [44] were selected as target organisms for marine environments. Soil nematodes [45], springtails [46], earthworms [47], and spiders [48] were selected as target organisms for soil environments. Neurotoxicity, reproductive toxicity, and developmental toxicity are highly complex and dependent on multiple molecular-signaling pathways. A comprehensive study of these three types of toxicity of pollutants is difficult. Acetylcholine receptors (AChR) are among the most commonly used receptors in neurotoxicity studies [49]; it is well established that the estrogen receptor plays an important role in the reproductive system [50] and thyroid hormone receptors are essential for normal development [51]. To facilitate simulations, these three key receptors were selected as target receptors to characterize the three types of toxic effects. The AChR, estrogen receptor, and thyroid hormone receptor were selected as target receptors for biological neurotoxicity [52], reproductive toxicity [53], and developmental toxicity [54], respectively.

Therefore, the present study focused on the toxic effects of TPs in various environmental media. Five common types of rubber—NR, SBR, BR, NBR, and IIR—were selected for analysis, and molecular docking and molecular dynamics were used to determine the different toxicities of each TP in each environment. The damage mechanism of the rubber monomer was identified on the basis of the bond interaction between the rubber monomer and the toxic target protein, and the harmful effect of TPs on food-chain organisms in various environmental media was comprehensively analyzed.

## 2. Materials and Methods

### 2.1. Sample Preparation and Characterization

TPs are often complex mixtures and tires are manufactured with a variety of additives in addition to the rubber itself. These additives can enter the environmental media with the TPs and have environmental impacts. Tire additives have been shown to be a major contributor to the toxicity of TPs [55]. For this reason, this study focuses on the environmental impact of rubber polymers and additives, leaving aside the environmental hazards of bitumen and heavy metals that may be carried by TPs. The present study selected five types of commonly used rubber with different rubber monomers for analysis—NR, SBR, BR, NBR, and IIR [56]. Rubbers with different additive characteristics, such as antioxidants, plasticizers, flame retardants, light and heat stabilizers, and lubricants, were selected [57]. The combinations of the five rubber monomers and their additives are shown in Table 1.

This paper investigates the biohormonal effects of TPs in various environmental media by referring to the relevant references and combining the PDB database, NCBI database, and UniProt database. It has been shown that specific receptors can be selected when studying the toxic effects of microplastics on organisms [58]. Therefore, the typical receptors of three kinds of hormone effects (neurotoxic, developmental toxicity, and reproductive toxicity) in each organism were selected as research objects and TPs were docked to the receptors to simulate their binding to hormone receptors and characterize the hormonal effects of TPs. The corresponding receptor numbers and sources for each species are shown in Table 2.

### 2.2. Characterization of Toxicity of TPs in Different Environmental Media—Molecular-Dynamics Approach

The molecular-docking approach is based on the lock-and-key principle, which is used for binding small ligand molecules to receptor proteins [59]. The space shape and energy between the ligand and acceptor during docking should be matched to obtain the most stable binding conformation [60]. Rubber is a polymer compound but its structure is not complex and is usually formed by specific structural units linked hundreds or thousands of times by covalent bonds. However, polymer structures do not facilitate molecular-dynamics calculations. Therefore, we used the idea of the article by Chen et al. to simplify the structure of the rubber 5-mer for the subsequent molecular-dynamics simulation study. In the present study, the neurotoxicity, reproductive toxicity, and developmental toxicity of five TPs, including NR, SBR, BR, NBR, and IIR, on the freshwater food chain, marine food chain, and soil-environment organisms were explored. The LibDock docking module of Discovery Studio 2020 software was used to dock the rubber monomer 5-mer to target proteins. Before docking, the protein molecules were pretreated to remove ligands and water from the proteins. A conformational search of the ligand molecule, followed by the analysis of the receptor’s binding site and generation of a series of available cavities, was carried out. These cavities were then matched energetically and geometrically to the available cavities of the protein to obtain a docked protein–ligand complex [61,62]. GROMACS software automates the processing of a wide range of biomolecules such as proteins, nucleic acids, and lipids and has all the usual force fields for these molecules built in, supporting a wide range of implicit solvent models and new free-energy algorithms, plus multi-threading for efficient parallelization [63]. GROMACS can be used to great advantage in the simulation of the dynamics of complex multi-molecular systems by increasing the efficiency of simulations for high-throughput and massively parallel simulations of polymers, crystals, and biomolecular solutions [64]. The complex system of the ligand molecules and proteins was transferred to GROMACS 4.6.5 software in the Dell PowerEdge R7525 server, and the dynamics of the molecular-docking complex were simulated under the force field of GROMOS96 43A1.

The protein–ligand complex was placed in a created cube box, and the shortest distance between the box boundary and the complex was 1 nm. Because rubber additives and rubber polymers coexist in nonbonding interactions, the additive was added to the simulation system as an external condition during the kinetic simulation [65,66]. SPC-type model water (SPC216) was added to the box. To ensure electroneutrality of the entire system, it was necessary to add an appropriate amount of Na^+^ or Cl^−^ to the box to balance the charge and replace a medium number of water molecules in the box. The molecular-dynamics simulation was divided into energy minimization, NVT temperature control, NPT pressure control, and MD equilibrium simulation. The energy was minimized by the steepest descent method. When the energy converged to 1000 kJ/mol, the system energy was in equilibrium. In the NVT and NPT, the temperature was set to 300 K, the pressure was set to 1 bar, and the protein–ligand complex was bound. The simulation step was set to 2 fs, and the number of simulation steps was set to 50,000. In the final MD equilibrium simulation, the position constraints were removed and the final simulation of the complex was performed under the original set temperature and pressure conditions. The simulation step size was 2 fs and the number of simulation steps was 100,000. Finally, the binding energy of the rubber monomer 5-mer to the toxic target protein was calculated by the Molecular Mechanics/Poisson–Boltzmann Surface Area (MMPBSA) equation, which requires the binding energy of the complex, protein, and ligand to be calculated separately. MMPBSA is an endpoint-binding free-energy calculation method with important applications in structure-based virtual screening with an accuracy of 1 kcal/mol. The equilibrium trajectories of the protein and ligand complexes were determined before calculation. The binding energy was calculated by the following formula [59]:(1)$$G_{bind}=G_{complex}-G_{free-protein}-G_{free-ligand}$$
where $G_{complex}$ is the binding energy of the protein–ligand complex, $G_{free-protein}$ is the binding energy of the protein, and $G_{free-ligand}$ is the binding energy of the ligand.

Binding energy of molecules in solution:(2)$$G=E_{gas}-TS_{gas}+G_{solvation}$$
where $E_{gas}$ is the gas-phase energy and entropy contributions, $TS_{gas}$ is the bond-angle interaction, and $G_{solvation}$ is the free energy of solvation, where the free energy of solvation can be decomposed into two components—the free energy of polar solvation ($G_{polar}$) and the free energy of nonpolar solvation ($G_{nonpolar}$):(3)$$G_{solvation}=G_{polar}+G_{nonpolar}$$

The gas-phase energy and entropy contributions were calculated according to the MM method:(4)$$E_{gas}=E_{MM}=E_{bond}+E_{angle}+E_{dihedral}+E_{vdw}+E_{coulomb}$$
where $E_{bond}$ is the bond interaction, $E_{angle}$ is the bond-angle interaction, $E_{dihedral}$ is the dihedral interaction, $E_{vdw}$ is the van der Waals interaction, and $E_{coulomb}$ is the Coulomb electrostatic interaction.

### 2.3. Comprehensive Toxicity Characterization of TPs in Different Environmental Media—Multilayer Empowerment Method

The numerical patterns of neural, developmental, and reproductive toxicities binding energies of five rubbers against several organisms were analyzed. A multilayer empowerment method was used to investigate the combined toxicity characteristics of five TPs in different environmental media. First, the toxicity values of five rubbers in different environmental media were analyzed to obtain three toxicity weights (first level) using a coupled-normalization method–principal-component analysis (PCA)–factor-analysis weighting method. Here, the normalization method was used for data standardization, and PCA combined with the factor-analysis weighting method was used for toxicity weighting. Second, the weights of four organisms (second level) and three environmental media (third level) were calculated separately using the subjective-weighting method. The above calculation process was completed using Python software. Normalization, PCA, and factor-analysis weighting were performed as follows.

(1) Data standardization—normalization method [67]
(5)$${\left\|X\right\|}_{2}=\sqrt{\sum_{i=1}^{n}x_{i}^{2}}$$
(6)$$X^{\prime }=\frac{X_{i}}{{\left\|X\right\|}_{2}}$$
where $X^{\prime }$ is the binding-energy value of normalized rubber and toxic receptor, $X_{i}$ is the original binding-energy value of rubber and toxic receptor, and ${\left\|X\right\|}_{2}$ is the L2 norm of the row vectors of the three toxicity numerical matrices.

(2) Feature-dimension reduction—PCA method [68]:(7)$$m=\left(\begin{array}{cccc} X_{11}^{\prime } & X_{12}^{\prime } & \cdots & X_{1n}^{\prime }\\ X_{21}^{\prime } & X_{22}^{\prime } & \cdots & X_{2n}^{\prime }\\ \vdots & \vdots & \ddots & \vdots \\ X_{m1}^{\prime } & X_{m2}^{\prime } & \cdots & X_{mn}^{\prime } \end{array}\right)=\left(\begin{array}{cccc} X_{1} & X_{2} & \cdots & X_{n} \end{array}\right)$$
(8)$$\left\{\begin{array}{c} F_{1}=a_{11}X_{1}+a_{21}X_{2}+\cdots a_{m1}X_{m}\\ F_{2}=a_{12}X_{1}+a_{22}X_{2}+\cdots a_{m2}X_{m}\\ \cdots \\ F_{n}=a_{1n}X_{1}+a_{2n}X_{2}+\cdots a_{mn}X_{m} \end{array}\right.$$
where $X=(X_{1},X_{2},\ldots ,X_{n})$ is the vector of normalized values of the binding energy of rubber to neurological-, developmental-, and reproductive-toxicity receptors; $F=(F_{1},F_{2},\ldots ,F_{n})$ is the vector of principal components for each toxicity; and $a_{mn}$ is the factor-loading coefficient, which is the eigenvector corresponding to the eigenvalues of the covariance matrix of X. In the present study, the decomposition module in the scikit-learn machine-learning library was used to build the PCA model, and the normalized TPs could reduce the dimension of toxicity in different environmental media. The specific code is shown in Appendix A.

(3) Determination of indicator weights—factor-analysis weighting method [69]:(9)$$b_{ik}=\frac{\left|a_{ik}\right|}{\sum_{j=1}^{n}\left|a_{ij}\right|},k=1,2,\ldots n;i=1,2,\ldots ,m$$
(10)$$c_{k}=\sum_{i=1}^{m}b_{ik},k=1,2,\ldots n$$
(11)$$W_{k}=\frac{c_{k}}{\sum_{j=1}^{p}c_{j}},k=1,2,\ldots n$$
where $b_{ik}$ is the evaluation of the principal-component vector $F_{i}$ for the three biotoxins to obtain the respective toxicity weights; $c_{k}$ is the sum of the weights of the normalized numerical vector X of the binding energy of rubber to neurological-, developmental-, and reproductive-toxicity receptors; and $W_{k}$ is the weight of the numerical vector X of the binding energy of rubber to neurological-, developmental-, and reproductive-toxicity receptors, calculated by the $c_{k}$ normalization method.

### 2.4. Characterization of the Toxicity of SBR in Different Environmental Media—Minimum-Value and Feature-Aggregation Methods

In the present study, the minimum-value method combined with the aggregation method of characteristics was used to study the numerical law of the toxicity of SBR in different environmental media, and the toxicity characteristics of SBR in different environmental media were analyzed. First, the toxicity values of three environmental media exposed to SBR were standardized by the minimum-value method; then, the dimension reduction of toxicity values was analyzed by the feature-aggregation method. The calculations of the minimum-value and feature-aggregation methods were as follows.

(1) Data normalization—minimum-value method [70]:(12)$$X^{\prime }=\frac{X}{X_{\min}}$$
where $X^{\prime }$ is the binding-energy value of rubber and a toxic receptor treated by the minimum-value method, X is the original binding-energy value of rubber and a toxic receptor, and $X_{\min}$ is the minimum original binding-energy value of rubber and a toxic receptor.

(2) Feature-dimension reduction—feature-aggregation method [71]:

Feature agglomeration refers to the simplification of multidimensional data by combining similar features. Here, we used the cluster module in the scikit-learn machine-learning library to construct a feature-aggregation model using the feature-agglomeration method in the cluster module. This was done to reduce the dimensionality of the toxicity values of SBR in different environmental media after processing by the minimum-value method. The model specifies a numerical-clustering number of 1 for each toxicity value in the three environmental media. The code is presented in Appendix A.

### 2.5. Characterization of Different Toxicities of SBR to Freshwater Environmental Organisms—Interval-Scaling and Entropy-Weighting Methods

The interval-scaling and entropy-weighting methods were used to process the toxicity values and analyze three toxicity features of SBR in freshwater environmental media. First, the interval-scaling method was used to standardize the toxicity values of SBR in freshwater environmental media; then, the entropy-weighting method was used to calculate the toxicity weights of the four types of organisms. The calculated and processed toxicity data were used for subsequent analysis of the results. The interval-scaling and entropy-weighting calculations were as follows.

(1) Data normalization—rescaling method [72]:(13)$$X_{\mathrm{std}}=\frac{X-X_{\min}}{X_{\max}-X_{\min}}$$
where X represents the raw binding-energy value of SBR and a toxic receptor, $X_{\min}$ represents the minimum binding-energy value of SBR and a toxic receptor, and $X_{\max}$ represents the maximum binding-energy value of SBR and a toxic receptor.

(2) Determination of indicator weights—entropy-weighting method [73]:(14)$$P_{ij}=X_{ij}/\sum_{i}^{m}X_{ij}$$
(15)$$e_{j}=-\frac{1}{\ln m}\sum_{i=1}^{m}p_{ji}\ln p_{ji}$$
(16)$$\omega_{j}=(1-e_{j})/\sum_{j=1}^{n}\left(1-e_{j}\right)$$
(17)$$a_{ji}=1+(X_{ji}-X_{j\min})\times /(X_{j\max}-X_{j\min})$$
(18)$$b_{i}=\sum_{j=1}^{3}\left(\omega_{j}\times a_{ji}\right)$$
where $P_{ij}$ is the weight of the effect of organism *i* in environmental medium *j*, $e_{j}$ is the entropy value of environmental medium *j*, $\omega_{j}$ is the entropy weight of environmental medium *j*, and $b_{i}$ is the combined toxicity of organism *i* in each environmental medium.

### 2.6. Characterization of the Toxicity of SBR to Different Freshwater Environmental Organisms—Standard-Deviation Normalization Method

To analyze the characteristics of the toxic effects of SBR on four organisms in the freshwater environment, the standard-deviation normalization method was used to analyze the numerical patterns of toxicity in freshwater environmental media exposed to SBR. The standard-deviation normalization formula was as follows [74]:(19)$$x=\frac{x-\bar{x}}{\sigma }$$
where $\bar{x}$ is the mean value of the toxicity of SBR to organisms in the freshwater environment, and σ is the standard deviation of the toxicity of SBR to organisms in the freshwater environment, with a mean value of 0 and a standard deviation of 1 for the processed data.

## 3. Results and Discussion

### 3.1. Toxicity Features of TPs in Different Environmental Media

Molecular-dynamics simulations were used to characterize the magnitude of toxicity of TPs in various environmental media. Binding-energy values for the neurological, developmental, and reproductive toxicity of NR, SBR, IIR, NBR, and NBR (and the additive components commonly used) to organisms in marine, freshwater, and soil environments were calculated [75,76]. For this purpose, we used molecular-dynamics simulations (Table 3) (the higher the absolute value of the toxicity in the table, the higher the toxicity of rubber to the organisms).

#### 3.1.1. Integrated-Toxicity Analysis of TPs Based on Multilevel-Empowerment Model

A multilevel-empowerment method was used to analyze the cumulative toxicity of the five TPs in three environmental media. By assigning three forms of toxicity to organisms in various environmental media, four types of biological organisms in various environments, and weights to three environmental media, the multilevel-empowerment model was constructed to determine the combined toxicity values of the five rubbers in various environmental media.

1. Characterization of different toxicity profiles in environmental media based on PCA and factor-analysis weighting method

PCA and the factor-analysis weighting method were used to establish the weights of the first-level indicators. To ensure the same order of magnitude for the various toxicity results, the normalization method was first used to dimensionlessly process the mean toxicity values (Table 3) of TPs in different environmental media.

Assessing the characteristics of the three toxicities of TPs in various environmental media was challenging because 60 dimensions of data were available for each of the three toxicities (Table 4). To determine the covariance eigenvalues and variance contributions of each component of the toxicity numerical matrix, the PCA model was built using machine-learning techniques to perform PCA on the standardized toxicity data (Table 5). Table 5 shows that reducing the features to one dimension may save 58.11% of the original data information, whereas reducing the features to two dimensions can preserve approximately 100% of the original data information. Only two principal-component characteristics were required to be recovered from the 60 data dimensions for each toxicity to characterize the original data information because the cumulative contribution of the eigenvalue satisfied the requirement to define the number of principal components of the PCA model (85% or more).

The two-dimensional main component features of the 60 data dimensions of the three toxicities were extracted using the PCA algorithm model. The retrieved two-dimensional features are displayed in Table 6. The weights of the three toxicities were determined using the factor-analysis weighting method by analyzing the eigenvalues of the two-dimensional main components. The computational procedure and outcomes are displayed in Table 6.

The weights for neurological, developmental, and reproductive toxicity were calculated as 36.6%, 30.3%, and 33.1%, respectively (Table 6). The TPs showed consistent impacts on the three types of toxicity in various environmental media. The toxicity values of TPs in various environmental media (Table 3) were multiplied by the three toxicity weights and then the three toxicity values for each organism were summed to determine the combined toxicity of TPs for each organism in the three types of environmental media (Table 7).

2. Toxicity characterization of different environmental-media organisms based on a subjective-empowerment model

The subjective-empowerment method was used to allocate the second-level indications. MPs are a potential threat to the environment in all environmental media. They have been found in aquatic ecosystems worldwide and have entered the food chain [58,77]. They can bioaccumulate and biomagnify in the food chain after entering it, which could affect the entire food chain [78,79]. There is a predatory relationship between organisms at all trophic levels in seawater and freshwater, and TPs are transferred from higher-trophic-level organisms to trophic-level organisms through biological food uptake, resulting in bioconcentration of TPs. Consequently, the four species were given weights of 10%, 20%, 30%, and 40%, respectively, in the trophic-level hierarchy for the marine and freshwater ecosystems. Because there are no predator–prey interactions among the three species of nematodes, hoppers, and earthworms that consume soil bacteria, fungi, and humus in the soil environment, TPs do not enter the organisms through the food chain [80]. Therefore, the three soil species were assigned weights of 10%, 20%, and 30%, respectively, considering that food consumption is often positively associated with body size, leading to a positive association between the amount of TPs swallowed by organisms and body size [81]. Since spiders are predators of soil organisms and have a higher trophic level than nematodes, hoppers, and earthworms [82], their weight was assigned as 40%. Using the above analysis, the weights for each organism in the different environmental media were calculated and multiplied by the toxicity values of the organisms exposed to TPs in different environmental media (Table 7). The toxicity values of TPs in the three environmental media are shown in Table 8.

3. Toxicity characterization of different environmental media based on a subjective-empowerment model

When a vehicle moves, the mechanical forces between the tire tread and the road surface cause the tires to wear out. This produces significant amounts of TPs that are deposited along the roadside and spread by wind, rain, gravity, ocean currents, and other environmental factors to freshwater, marine, and soil environments as well as the atmosphere [6,7,27]. Unice et al. [83] used an integrated watershed-scale-modeling approach to simulate the transport fate of tire-wear particulate matter in the Seine River basin in France. Model simulations revealed that only 2% of the particulate matter was exported to the ocean from the estuary and 18% was discharged to freshwater. The migration of TPs in the three aforementioned environmental media and the content of TPs in each environment were considered. Afterward, a subjective-empowerment approach was used to assign weights to the three environmental media at the third level of the index, with marine, freshwater, and soil environmental media set at 20%, 50%, and 30%, respectively.

The total toxicity of the five rubbers to the three environmental media was calculated to be −129.525 ± 10.406, −138.080 ± 13.142, −119.347 ± 8.974, −91.547 ± 9.812, and −120.276 ± 9.856 kJ/mol, respectively, by multiplying the above weights by the three environmental-toxicity values in Table 8. SBR had the highest cumulative toxicity in the three environmental media, followed by NR. The two most common rubbers used in tires are SBR and NR, and SBR is the most abundant synthetic rubber [6,83]. SBR contains benzene, a heterocyclic compound with a complex breakdown behavior [84,85]. Consequently, SBR from tire wear poses a bigger threat to the environment, and the results of the present study also showed that SBR was the most toxic in the three environmental media.

#### 3.1.2. Toxicity Characterization of SBR in Different Environmental Media

The toxicity characteristics of SBR in three environmental media were analyzed using the minimum-value method coupled with the feature-aggregation method. The three toxicity features of SBR in the freshwater environment were explored using the interval-deflation method coupled with the entropy-weighting method.

1. Characterization of SBR toxicity in different environmental media based on the feature-aggregation model

The toxicity values for species exposed to SBR in various environmental media differed greatly (Table 3). In the present study, the data were preprocessed using the minimum-value method, and the preprocessed derived dimensionless values are shown in Table 9.

The toxicity in each environmental medium had 12 dimensional features (Table 9), which made it difficult to visually analyze the effects of SBR in the three environmental media. The feature-aggregation method was used to reduce the dimensionality and analyze the toxicity values of SBR. Machine learning was used to build a feature-aggregation model that reduced the 12-dimensional toxicity features to one-dimensional features. The model determined toxicity-feature values of 1.502, 1.688, and 1.544 for each environmental medium, respectively. The rank order of the toxicity of SBR was the same as that of the toxicity of TPs in the three environmental media, as determined by the multilayer-weighting method. Our results showed that organisms in freshwater environments are more vulnerable to TPs.

2. Characterization of the toxicity of SBR to different organisms in the freshwater environment based on the entropy-weighting model

Three toxicity traits of freshwater environmental organisms under exposure to SBR were examined using the entropy-weighting method because freshwater environmental organisms are more susceptible to SBR toxicity (Table 3). The rescaling method was used to standardize the toxicity values of SBR in the freshwater environment. The entropy-weighting method was then used to calculate the weights of the four organisms. The standardized values and the outcomes of the entropy-weighting method are displayed in Table 10.

The combined values of neurotoxicity, developmental toxicity, and reproductive toxicity of SBR to freshwater environmental organisms were calculated to be −168.260 ± 15.270, −133.529 ± 12.521, and −145.112 ± 12.468 kJ/mol, respectively, by multiplying the weights of the four organisms (Table 10) by their toxicity values (Table 3) and adding them up. The rank order of the toxicity of the three organisms was the same as that of the three toxicity weights determined using PCA and factor-analysis weighting in the multilayer-empowerment model. Similarly, the rank order of the three biological toxins showed the same pattern as the three toxicity weights calculated by PCA and the factor-analysis weighting method in the multilayer-empowerment model of this paper.

### 3.2. Mechanism Analysis of Differences in Toxicity of TPs

The standard-deviation normalization method was used to process the freshwater environmental-toxicity values without a dimension (Table 3) under SBR exposure, and the results are shown in Table 11 to visually analyze the variations in the toxicity features of SBR in freshwater environments.

The three toxicity values of SBR were positive for shrimp and snails but negative for freshwater nematodes and freshwater fish (Table 11). Thus, SBR was substantially less hazardous to freshwater nematodes and freshwater fish than to snails and shrimp. TPs tend to accumulate more easily in crustaceans than in insects or fish [86]. After entering the body of fish, MPs can be enriched in the digestive system, and most of them can be eliminated via the fish’s metabolism. In contrast, because crustaceans are filter feeders and prone to accumulating MPs, the harmful effects of SBR on snails and shrimp were more apparent. In addition, because shellfish and mollusks are eaten whole, they are more likely to cause hazardous effects to human health [87]. This is in line with the distinctions in freshwater toxicity mentioned above.

Studies have shown that MPs readily interact with cell membranes due to their hydrophobic nature and acquire favorable contacts with hydrophobic lipid tails, thus leading to MPs directly piercing through the membrane, then interacting with proteins in the cell [88]. The 2D distribution of important amino-acid residues of the AChR that binds to the 5-polymer SBR was mapped using freshwater nematodes, snails, shrimp, and freshwater fish, respectively, using the amino-acid residue-analysis tool of the Discovery Studio 2020 software (Figure 1). In all four organisms living in freshwater environments, van der Waals interactions predominated during the binding of AChR to SBR. The following amino acids were involved in the interaction: glutamic acid (GLU: hydrophilic), lysine (LYS: hydrophilic), phenylalanine (PHE: hydrophobic), threonine (THR: hydrophilic), isoleucine (ILE: hydrophobic), serine (SER: hydrophilic), glutamine (GLN: hydrophilic), methionine (MET: hydrophobic), glycine (GLY: hydrophobic), proline (PRO: hydrophobic), alanine (ALA: hydrophobic), valine (VAL: hydrophobic), aspartic acid (ASP: hydrophilic), tryptophan (TRP: hydrophobic), tyrosine (TYR: hydrophobic), histidine (HIS: hydrophilic), asparagine (ASN: hydrophilic), arginine (ARG: hydrophilic), and leucine (LEU: hydrophobic). During the binding of freshwater nematodes AChR to SBR, 18 amino-acid residues were involved in van der Waals interactions and 11 of those residues (61.11%) were hydrophobic (Figure 1a). When the snail AChR bound to SBR, 19 amino-acid residues were involved in van der Waals interactions (Figure 1b), of which 11 hydrophobic amino-acid residues made up 57.89% of the total. The binding of shrimp AChR to SBR involved 32 amino-acid residues, 23 of which were hydrophobic and accounted for 71.88% of the total number of van der Waals-interacting amino-acid residues (Figure 1c). When the freshwater-fish AChR bound to SBR, 13 amino-acid residues were involved in van der Waals interactions, and 8 were hydrophobic, accounting for 61.54% of the total (Figure 1d). Consequently, hydrophobic amino acids accounted for a large portion of SBR’s affinity for AChR and van der Waals interactions between molecules were responsible for producing hydrophobic interactions.

Hydrogen bonding and electrostatic, polar, hydrophobic, and nonbonding interactions such as van der Waals dominate the binding of MPs to receptors [89]. Van der Waals interactions, which can affect the stability of binding between ligands and receptors, are dominant in the binding process of MPs to receptors [90]. In contrast, π-π interactions offer stability that correlates with ligand–receptor-binding stability [91]. Hydrophobic interactions are primarily responsible for the metabolism of exogenous organic molecules [92]. The stability of the binding between ligands and receptors is determined by van der Waals and hydrophobic interactions, according to the analysis of important amino-acid residues in the binding of SBR to AChR. The results of the present study are consistent with those of previous studies.

Compared with the AChR key amino-acid-residue map of freshwater nematodes (Figure 1a), the residue map of snails (Figure 1b) contained an identical number of hydrophobic amino-acid residues, with the same hydrophobic forces but with more amino-acid residues that reflected van der Waals interactions and contained π-π and π-sigma bonds. The residue map of shrimp (Figure 1c) showed 109.09% more hydrophobic amino-acid residues, 77.78% more residues that represented van der Waals contacts, and π-π bonds. Compared with the AChR key amino-acid-residue map of freshwater fish (Figure 1d), the residue map of snails (Figure 1b) included 10% more hydrophobic amino-acid residues (Figure 1d). Furthermore, the amino-acid residues with van der Waals interactions were more evident, whereas π-π interactions were more potent. Although the residue map of freshwater fish (Figure 1d) had more π-π and π-sigma bonds than the residue map of shrimp (Figure 1c), the residue map of shrimp (Figure 1c) had 130% more hydrophobic amino-acid residues and 113.33% more van der Waals-interacting amino-acid residues. In addition, the shrimp-residue map (Figure 1c) had 230% more hydrophobic amino-acid residues and 213% more van der Waals-interacting amino-acid residues than the fish-residue map (Figure 1d).

In conclusion, the AChR complexes of SBR with snails and shrimp were more stable than those with freshwater nematodes and freshwater fish due to the stronger van der Waals, hydrophobic, π-π, and π-sigma interactions. Furthermore, snails and shrimp were more vulnerable to the neurotoxic effects of SBR than freshwater nematodes and freshwater fish. Thus, future research should focus on these effects.

## 4. Conclusions

The toxicity of TPs in various environmental media was determined using molecular-dynamics simulation methods and was represented by binding-energy values. Different mathematical techniques were then used to extract information from toxicity values and identify the toxicity characteristics of TPs. The results were as follows:

(1) The combined toxicity of the five TPs was investigated using a multilevel-weighting model. The results of the toxicity-category weights (first level) generated using data-normalization, PCA, and factor-analysis weighting methods showed that the organisms were more susceptible to neurotoxicity caused by TPs. The subjective-empowerment method for organisms in various environmental media (second level) revealed that the impact of TPs on various organisms increased with the trophic level. The TPs had the greatest effect on organisms in the freshwater environment, according to the subjective-empowerment method for the three environmental media (third level). SBR was the most harmful in various environmental media, according to the multilevel-weighting model’s analysis of the comprehensive toxicity of the five TPs.

(2) Using a combination of the minimum-value and characteristic-aggregation methods, the effects of SBR were examined in three environmental media and the findings revealed that SBR had the most pronounced toxicity profile in the freshwater environment. Using the interval-deflation method coupled with the entropy-weighting method, the various toxicities of SBR to freshwater organisms were calculated, with neurotoxicity having the greatest impact.

(3) Using standard-deviation standardization, it was discovered that among the organisms in the freshwater environment, snails and shrimp were the most vulnerable to SBR due to biological neurotoxicity being more sensitive to TPs. Amino-acid-residue analysis found that the SBR pentamers were more stable in complexes formed by binding to the AChR of snails and shrimp than to the AChR of nematodes and salmon. The present study provides theoretical support to identify and assess the hazardous environmental effects of TPs and a technical reference for further control and mitigation of the ecological effects of TPs on the aquatic environment.

## Figures and Tables

**Figure 1 polymers-15-01719-f001:**
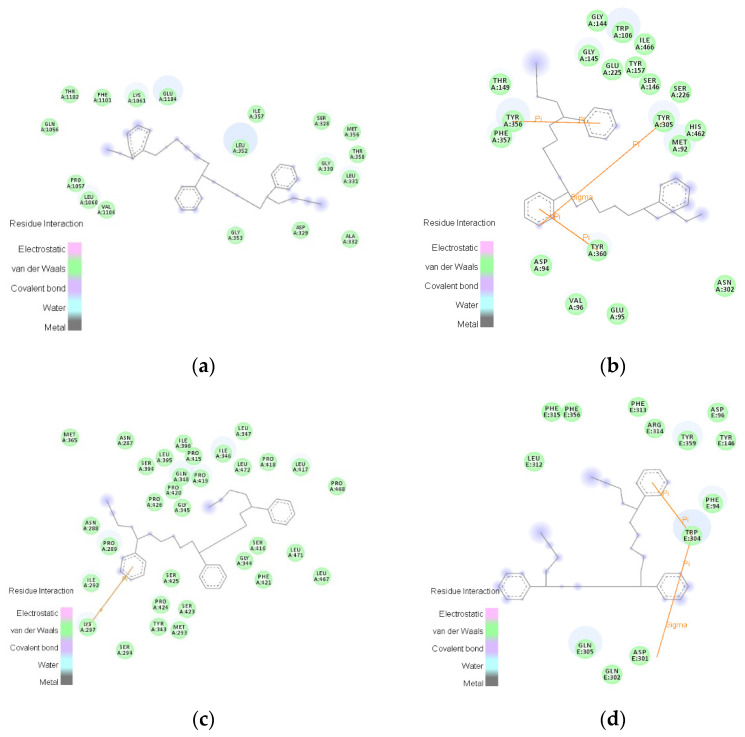
Two-dimensional distribution of key amino-acid residues (green block shows van der Waals interactions). (**a**) Freshwater nematodes; (**b**) snails; (**c**) shrimp; (**d**) freshwater fish.

**Table 1 polymers-15-01719-t001:** Main rubber and additive combinations.

Rubber	Antioxidant	Plasticizer	Flame Retardant	Light Stabilizer	Heat Stabilizer	Lubricant
NR	DTBHQ	Naphthenic oil	Triphenyl phosphate	Tinuvin 326	Zinc stearate	Pentaerythrityl tetrastearate
SBR	TMQ	Naphthenic oil	Decabromodiphenyl oxide	Tinuvin 770	PA6	N,N′-ethylenedi (stearamide)
BR	Antioxidant -1520	DBP	Decabromodiphenyl oxide	UV-531	4,4-bis (α, α-dimethylbenzyl) diphenylamine	N,N′-ethylenebisoleamide
NBR	Antioxidant -2246	Diisooctyl adipate	Bisphenol A	Tinuvin 765	Triphenyl phosphite	Oleamide
IIR	Antioxidant-BHT	Chlorinated paraffins	Tris(1-Chloro-2-Propyl) Phosphate	Tinuvin 622	Calcium stearate	Propanal

**Table 2 polymers-15-01719-t002:** Toxic-receptor numbers and sources for different environmental-media organisms.

Environmental Media	Species	Neurotoxic-Receptor Number and Source (Acetylcholine)		Developmental-Toxicity-Receptor Number and Source (Thyroid Hormone)		Reproductive-Toxicity-Receptor Number and Source (Estrogen)	
Freshwater environment	Freshwater nematodes	A0A2A6D033	UniProt	A0A2A6BEU0	UniProt	A0A2A6CC73	UniProt
	Snails	A0A8F1SZL8	UniProt	A0A182YVU7	UniProt	W8SMZ8	UniProt
	Shrimp	A0A3R7MHH7	UniProt	A0A3R7QNK1	UniProt	A0A423TCB3	UniProt
	Freshwater fish	BAH11081.1	NCBI	Q766D2	NCBI	P50241	UniProt
Marine environment	Marine nematodes	A0A0U2IDR0	UniProt	D2XNK4	NCBI	B5THM8	UniProt
	Mussels	A0A8B6HK24	UniProt	A0A4Y5MV01	UniProt	A0A8B6H7D5	UniProt
	Crabs	A0A6B9F3J7	UniProt	MPC23433.1	NCBI	A0A097C257	UniProt
	Marine fish	2ACE	PDB	Q9W6I9	NCBI	P16058	UniProt
Soil environment	Soil nematodes	Q65XS8	UniProt	F11A1.3a	NCBI	T01B10.4a	NCBI
	Springtails	A0A1D2NK24	UniProt	A0A1D2N3W7	UniProt	A0A1D2NKF5	UniProt
	Earthworms	Q34946	UniProt	A0A2I7YV05	UniProt	QCI03097.1	NCBI
	Spiders	A0A4Y2A8I0	UniProt	A0A2L2YDA1	UniProt	A0A087U0Z5	UniProt

**Table 3 polymers-15-01719-t003:** Toxicity values of TPs in different environmental media (kJ/mol).

Environmental Media	Species	Toxic Types	Rubber Types									
			NR		SBR		IIR		NBR		BR	
			Mean	Std	Mean	Std	Mean	Std	Mean	Std	Mean	Std
Marine environment	Marine nematodes	Neurotoxic	−85.814	±11.435	−134.724	±13.936	−103.666	±13.190	−56.265	±13.899	−109.635	±10.219
		Developmental toxicity	−190.042	±9.203	−114.182	±23.364	−157.660	±5.620	−130.506	±11.160	−125.605	±8.270
		Reproductive toxicity	−127.508	±12.039	−136.188	±9.322	−107.039	±8.367	−49.666	±8.429	−80.121	±14.786
	Mussels	Neurotoxic	−120.140	±14.242	−105.094	±11.984	−102.452	±10.648	−85.575	±6.831	−60.038	±8.120
		Developmental toxicity	−109.222	±7.781	−84.286	±15.886	−83.981	±11.494	−68.699	±41.885	−137.881	±10.332
		Reproductive toxicity	−99.354	±11.658	−130.416	±17.688	−129.412	±18.393	−66.284	±11.621	−120.806	±6.779
	Crabs	Neurotoxic	−134.735	±19.536	−138.866	±10.444	−103.575	±13.660	−42.803	±14.144	−153.636	±15.710
		Developmental toxicity	−103.470	±9.381	−119.700	±26.069	−97.682	±10.828	−82.711	±10.619	−92.112	±6.798
		Reproductive toxicity	−102.930	±11.120	−100.485	±18.488	−107.063	±6.739	−58.793	±11.303	−61.894	±19.045
	Marine fish	Neurotoxic	−143.640	±9.761	−227.859	±16.971	−130.112	±10.762	−80.107	±13.503	−121.493	±11.758
		Developmental toxicity	−170.626	±9.586	−97.698	±7.362	−130.174	±8.017	−116.101	±7.980	−107.314	±11.222
		Reproductive toxicity	−150.872	±6.363	−130.131	±13.778	−86.038	±6.615	−23.478	±3.563	−74.073	±10.225
Freshwater environment	Freshwater nematodes	Neurotoxic	−73.776	±21.276	−102.638	±14.695	−88.644	±8.896	−42.805	±11.989	−62.687	±8.654
		Developmental toxicity	−148.863	±12.367	−85.244	±20.565	−149.897	±6.213	−118.048	±9.788	−80.690	±15.844
		Reproductive toxicity	−132.451	±7.113	−122.946	±15.471	−104.044	±10.566	−35.948	±9.672	−85.521	±9.082
	Snails	Neurotoxic	−175.723	±9.204	−209.514	±18.150	−130.063	±12.114	−107.159	±5.265	−145.993	±12.572
		Developmental toxicity	−138.244	±12.237	−157.761	±8.950	−81.125	±7.704	−70.769	±7.796	−139.429	±12.300
		Reproductive toxicity	−99.302	±10.569	−156.511	±12.364	−165.297	±10.441	−110.16	±6.964	−176.979	±8.330
	Shrimp	Neurotoxic	−134.723	±8.166	−214.905	±13.311	−112.634	±7.695	−92.276	±12.793	−105.606	±12.873
		Developmental toxicity	−136.396	±11.802	−159.702	±11.972	−105.391	±9.955	−121.405	±9.431	−135.354	±8.712
		Reproductive toxicity	−152.165	±9.279	−193.319	±7.826	−155.344	±5.892	−127.642	±5.432	−149.849	±6.892
	Freshwater fish	Neurotoxic	−142.027	±9.472	−101.777	±11.450	−132.187	±10.865	−142.093	±9.638	−145.136	±5.879
		Developmental toxicity	−129.466	±6.924	−105.937	±12.416	−134.523	±4.840	−141.062	±9.207	−139.252	±7.332
		Reproductive toxicity	−126.803	±19.028	−96.850	±14.008	−160.789	±7.737	−115.327	±12.091	−146.603	±7.914
Soil environment	Soil nematodes	Neurotoxic	−140.318	±12.305	−142.906	±15.985	−119.421	±9.168	−69.341	±15.983	−119.106	±11.630
		Developmental toxicity	−156.208	±12.194	−103.652	±12.223	−127.39	±6.012	−91.232	±11.483	−141.282	±10.433
		Reproductive toxicity	−97.488	±9.695	−132.416	±8.421	−88.463	±7.693	−42.997	±10.314	−76.235	±6.726
	Springtails	Neurotoxic	−144.001	±8.558	−120.260	±10.905	−127.982	±7.483	−74.308	±8.581	−117.241	±6.535
		Developmental toxicity	−81.106	±15.388	−108.633	±19.151	−64.389	±12.405	−40.046	±15.481	−94.386	±9.940
		Reproductive toxicity	−158.757	±6.618	−167.882	±11.916	−132.001	±9.169	−42.883	±5.863	−73.710	±5.269
	Earthworms	Neurotoxic	−166.664	±7.439	−191.017	±8.843	−120.004	±4.587	−100.327	±10.718	−134.435	±10.697
		Developmental toxicity	−100.351	±8.859	−90.403	±15.463	−89.585	±12.078	−28.353	±3.654	−79.751	±15.741
		Reproductive toxicity	−112.308	±11.928	−119.241	±9.449	−62.444	±21.694	−73.724	±15.030	−67.075	±28.081
	Spiders	Neurotoxic	−140.385	±8.686	−170.018	±13.283	−103.673	±6.816	−71.195	±11.000	−113.221	±6.736
		Developmental toxicity	−74.140	±12.342	−113.069	±16.365	−95.061	±10.149	−39.851	±8.307	−101.452	±12.796
		Reproductive toxicity	−91.384	±8.882	−102.088	±19.800	−140.865	±9.881	−114.247	±10.824	−148.510	±6.519

**Table 4 polymers-15-01719-t004:** Normalized toxicity values of TPs in different environmental media.

Environmental Media	Species	Toxic Type	Rubber Types				
			NR	SBR	IIR	NBR	BR
Marine environment	Marine nematodes	Neurotoxic	−0.088	−0.138	−0.106	−0.058	−0.112
		Developmental toxicity	−0.213	−0.128	−0.177	−0.147	−0.141
		Reproductive toxicity	−0.141	−0.151	−0.119	−0.055	−0.089
	Mussels	Neurotoxic	−0.123	−0.108	−0.105	−0.088	−0.062
		Developmental toxicity	−0.123	−0.095	−0.094	−0.077	−0.155
		Reproductive toxicity	−0.110	−0.145	−0.143	−0.073	−0.134
	Crabs	Neurotoxic	−0.138	−0.142	−0.106	−0.044	−0.158
		Developmental toxicity	−0.116	−0.134	−0.110	−0.093	−0.103
		Reproductive toxicity	−0.114	−0.111	−0.119	−0.065	−0.069
	Marine fish	Neurotoxic	−0.147	−0.234	−0.133	−0.082	−0.125
		Developmental toxicity	−0.192	−0.110	−0.146	−0.130	−0.120
		Reproductive toxicity	−0.167	−0.144	−0.095	−0.053	−0.082
Freshwater environment	Freshwater nematodes	Neurotoxic	−0.076	−0.105	−0.091	−0.044	−0.064
		Developmental toxicity	−0.167	−0.096	−0.168	−0.133	−0.091
		Reproductive toxicity	−0.147	−0.136	−0.115	−0.040	−0.095
	Snails	Neurotoxic	−0.180	−0.215	−0.133	−0.110	−0.150
		Developmental toxicity	−0.155	−0.177	−0.091	−0.079	−0.157
		Reproductive toxicity	−0.110	−0.173	−0.183	−0.122	−0.196
	Shrimp	Neurotoxic	−0.138	−0.220	−0.116	−0.095	−0.108
		Developmental toxicity	−0.153	−0.179	−0.118	−0.136	−0.152
		Reproductive toxicity	−0.169	−0.214	−0.172	−0.141	−0.166
	Freshwater fish	Neurotoxic	−0.146	−0.104	−0.136	−0.146	−0.149
		Developmental toxicity	−0.145	−0.119	−0.151	−0.158	−0.156
		Reproductive toxicity	−0.141	−0.107	−0.178	−0.128	−0.163
Soil environment	Soil nematodes	Neurotoxic	−0.144	−0.147	−0.122	−0.071	−0.122
		Developmental toxicity	−0.175	−0.116	−0.143	−0.102	−0.159
		Reproductive toxicity	−0.108	−0.147	−0.098	−0.048	−0.085
	Springtails	Neurotoxic	−0.148	−0.123	−0.131	−0.076	−0.120
		Developmental toxicity	−0.091	−0.122	−0.072	−0.045	−0.106
		Reproductive toxicity	−0.176	−0.186	−0.146	−0.048	−0.082
	Earthworms	Neurotoxic	−0.171	−0.196	−0.123	−0.103	−0.138
		Developmental toxicity	−0.113	−0.101	−0.101	−0.032	−0.090
		Reproductive toxicity	−0.124	−0.132	−0.069	−0.082	−0.074
	Spiders	Neurotoxic	−0.144	−0.174	−0.106	−0.073	−0.116
		Developmental toxicity	−0.083	−0.127	−0.107	−0.045	−0.114
		Reproductive toxicity	−0.101	−0.113	−0.156	−0.127	−0.165

**Table 5 polymers-15-01719-t005:** Eigenvalues and variance contributions of the components of the toxicity matrix under exposure to TPs in different environmental media.

Components	Initial Features			Extract Features		
	Eigenvalue	Variance Contribution	Cumulative Contribution	Eigenvalue	Variance Contribution	Cumulative Contribution
1	0.058	58.11%	58.11%	0.058	58.11%	58.11%
2	0.041	41.88%	99.99%	0.041	41.88%	99.99%
3	1.441 × 10^−31^	1.44 × 10^−29^%	100%			

**Table 6 polymers-15-01719-t006:** Calculated results of the toxicity features and weights of each organism under exposure to TPs in different environmental media.

Toxic Type	Two-Dimensional Features		Features Weights		ck	Weights
	ai1	ai2	bi1	bi2		
Neurotoxic	−0.144	−0.138	0.339	0.394	0.733	0.366
Developmental toxicity	0.213	−0.037	0.500	0.106	0.606	0.303
Reproductive toxicity	−0.069	0.175	0.161	0.500	0.661	0.331

**Table 7 polymers-15-01719-t007:** Toxicity values of TPs in different environmental media (kJ/mol).

Environmental Media	Species	Rubber Types									
		NR		SBR		IIR		NBR		BR	
		Mean	Std	Mean	Std	Mean	Std	Mean	Std	Mean	Std
Marine environment	Marine nematodes	−131.163	±10.616	−128.988	±14.847	−121.131	±8.904	−76.563	±10.842	−104.710	±10.834
	Mussels	−109.960	±11.002	−107.168	±14.695	−105.775	±13.148	−74.085	±18.833	−103.706	±8.103
	Crabs	−114.749	±13.087	−120.369	±17.528	−102.944	±10.101	−60.175	±11.711	−104.665	±13.642
	Marine fish	−154.203	±8.290	−156.125	±12.493	−115.554	±8.235	−80.411	±8.134	−101.517	±10.735
Freshwater environment	Freshwater nematodes	−115.918	±13.250	−104.087	±16.290	−112.285	±8.369	−63.321	±10.196	−75.690	±10.715
	Snails	−139.100	±10.299	−176.314	±12.903	−126.897	±9.861	−97.133	±6.436	−154.253	±10.708
	Shrimp	−140.998	±9.391	−191.051	±10.690	−124.566	±7.552	−112.793	±8.954	−129.246	±9.246
	Freshwater fish	−133.189	±11.579	−101.407	±12.246	−142.354	±7.678	−132.929	±10.030	−143.839	±6.816
Soil environment	Soil nematodes	−130.965	±11.038	−127.551	±11.862	−111.595	±7.448	−67.257	±12.264	−111.642	±9.295
	Springtails	−129.837	±9.729	−132.489	±13.411	−110.055	±9.308	−53.540	±9.515	−95.924	±6.952
	Earthworms	−128.608	±9.132	−136.813	±10.784	−91.757	±12.382	−69.735	±9.683	−95.599	±17.659
	Spiders	−104.120	±9.598	−130.308	±15.975	−113.366	±8.636	−75.942	±9.796	−121.328	±8.298

**Table 8 polymers-15-01719-t008:** Biological-toxicity values of 3 environmental media exposed to TPs (kJ/mol).

Environmental Media	Rubber Types									
	NR		SBR		IIR		NBR		BR	
	Mean	Std	Mean	Std	Mean	Std	Mean	Std	Mean	Std
Marine environment	−131.214	±10.504	−132.893	±14.679	−110.373	±9.844	−72.690	±11.618	−103.219	±11.091
Freshwater environment	−134.987	±10.834	−143.550	±12.315	−130.919	±8.146	−112.768	±9.005	−134.729	±8.714
Soil environment	−119.294	±9.628	−132.420	±13.494	−106.044	±9.775	−68.731	±9.953	−107.560	±10.937

**Table 9 polymers-15-01719-t009:** Standardized values for toxicity of SBR in different environmental media.

Environmental Media	Species	Toxicity		
		Neurotoxic	Developmental Toxicity	Reproductive Toxicity
Marine environment	Marine nematodes	1.598	1.355	1.616
	Mussels	1.247	1.000	1.547
	Crabs	1.648	1.420	1.192
	Marine fish	2.703	1.159	1.544
Freshwater environment	Freshwater nematodes	1.218	1.011	1.459
	Snails	2.486	1.872	1.857
	Shrimp	2.550	1.895	2.294
	Freshwater fish	1.208	1.257	1.149
Soil environment	Soil nematodes	1.695	1.230	1.571
	Springtails	1.427	1.289	1.992
	Earthworms	2.266	1.073	1.415
	Spiders	2.017	1.341	1.211

**Table 10 polymers-15-01719-t010:** Standardized values for toxicity of SBR in the freshwater environment and results of the entropy-weighting method.

Species	Toxic Type			Weights
	Neurotoxic	Developmental Toxicity	Reproductive Toxicity	
Freshwater nematodes	0.461	0.000	1.000	0.202
Snails	1.000	0.024	0.000	0.421
Shrimp	1.000	0.000	0.609	0.185
Freshwater fish	0.542	1.000	0.000	0.192

**Table 11 polymers-15-01719-t011:** Standardized values for toxicity of SBR in the freshwater environment.

Species	Toxic Type		
	Neurotoxic	Developmental Toxicity	Reproductive Toxicity
Freshwater nematodes	−0.992	−1.293	−0.537
Snails	0.950	0.944	0.390
Shrimp	1.048	1.004	1.406
Freshwater fish	−1.007	−0.655	−1.258

## Data Availability

All data generated and analyzed during this study are included in this article and Appendix A.

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
