# Peer review of "Exploring the Potential Hormonal Effects of Tire Polymers (TPs) on Different Species Based on a Theoretical Computational Approach"

_polymers, 2023, doi:10.3390/polym15071719_

Round 1

Reviewer 1 Report

Yu Wang, Hao Yang, Wei He, Peixuan Sun, Wenjin Zhao and Miao Liu “Exploring the Potential Hormonal Effects of Tire Polymers (TPs) on Different Species based on a Theoretical Computational Approach”, this article is a rather interesting work that uses appropriate approaches to study the toxicity of TPs with the help of molecular dynamics simulations, which may influence the future development strategy of recycling and give an additional incentive. However, before any decision is made on its publication, mandatory revision is required in order to clarify some points and increase its attractiveness to the general public journal polymers. See comments below.

1)      I recommend the authors to supplement the introduction. Since this study is aimed at determining the toxicity of the environment, I would also like to see a mention of the methods of recycling tire polymers. (for example, https://doi.org/10.3390/su14084503, https://doi.org/10.3390/polym13152494, https://doi.org/10.3390/app11177824).

2)      I recommend, in the methodology, to explain why this particular GROMACS 4.6.5 software was chosen for calculations. After all, there are many software packages for solving tasks, for example NAMD, CHARMM, AMBER.

3)      In equations (1-3), please explain (G_complex?, T?, S_gas) ?

Reviewer 2 Report

The manuscript dealt with the exploration of the potential Potential Hormonal Effects of Tire Polymers on Different Species. The manuscript is a well-written manuscript with timely needed information and data. Some minor corrections can be conducted before accepting the manuscript for publication.

  1. Section 2.1 data source: I have found that the authors have collected data from any source. As I have seen, the section is to sample preparation and characterization. It suggests to revise the section title.

2. Section 2.1, Page 4, line 140-163: It seems like a literature review.  Please revise or move this paragraph to section 1.  The authors may summarize how they conducted hormone analyses.

3. Experimental data present in Table 3 and Table 4 are the single experimental run. However, there is always doubt on the accuracy when data is presented from a single experimental run. It suggests providing experimental data from the double or triplicate experimental runs and presenting the data as mean value ± standard deviations. 
